# TOWARDS IMAGE COMPRESSION WITH PERFECT REALISM AT ULTRA-LOW BITRATES

**Marlène Careil**[1,2]**, Matthew J. Muckley**[1]**, Jakob Verbeek**[1]**, Stéphane Lathuilière**[2]
[1]Meta AI
[2]LTCI, Télécom Paris, IP Paris
{marlenec,mmuckley,jjverbeek}@meta.com
stephane.lathuiliere@telecom-paris.fr

## ABSTRACT

Image codecs are typically optimized to trade-off bitrate *vs*. distortion metrics. At low bitrates, this leads to compression artefacts which are easily perceptible, even when training with perceptual or adversarial losses. To improve image quality and remove dependency on the bitrate we propose to decode with iterative diffusion models. We condition the decoding process on a vector-quantized image representation, as well as a global image description to provide additional context. We dub our model "PerCo" for "perceptual compression", and compare it to state-of-the-art codecs at rates from 0.1 down to 0.003 bits per pixel. The latter rate is more than an order of magnitude smaller than those considered in most prior work, compressing a $512 \times 768$ Kodak image with less than 153 bytes. Despite this ultra-low bitrate, our approach maintains the ability to reconstruct realistic images. We find that our model leads to reconstructions with state-of-the-art visual quality as measured by FID and KID. As predicted by rate-distortion-perception theory, visual quality is less dependent on the bitrate than previous methods.

## 1 INTRODUCTION

Traditional image and video codecs are optimized for the rate-distortion function (Shannon, 1948), which minimizes the expected size of the data under a distortion constraint such as mean-squared error. Recent research has developed neural image compression methods that surpass handcrafted image compression codecs in terms of rate-distortion performance (Ballé et al., 2017; Ballé et al., 2018; Cheng et al., 2020; He et al., 2022a). However, optimization for the rate-distortion function comes at a cost of "realism", where realism is mathematically related to the statistical fidelity or $f$-divergence between the compressed image distribution and the true image distribution (Blau & Michaeli, 2018; 2019). The typical qualitative manifestation of unrealistic images is blurring.

Generative modeling compensates for artefacts such as blurring by introducing a divergence term (Agustsson et al., 2019; Mentzer et al., 2020; Muckley et al., 2023), typically in the form of an adversarial discriminator loss, which improves human-perceptual performance (Mentzer et al., 2020). Such codecs are called "generative compression" codecs, and are evaluated in terms of the rate-distortion-realism tradeoff (Blau & Michaeli, 2019). A further result of rate-distortion-realism theory is the theoretical possibility of a perfect realism codec, *i.e.*, a codec with no $f$-divergence, and zero FID across all rates, with no more than twofold increase in mean-squared error from the rate-distortion optimal codec. This result motivates research on perfect realism codecs, where original and reconstruction are different from each other, but it is not possible to tell which is which. Such codecs are particularly interesting for extremely low bitrate settings, where existing codecs introduce severe artefacts that are easily perceptible, see Fig. 1. So far, the primary effort towards building such a codec was the work of Theis et al. (2022), who demonstrated lower FID scores than HiFiC (Mentzer et al., 2020) via a combination of a pretrained variational diffusion model and reverse channel coding. However, computational constraints prohibited application to images larger than $64 \times 64$ pixels.

In this paper we make further progress towards perfect-realism codecs. Similar to Theis et al. (2022), we leverage a pretrained text-to-image diffusion model; however, rather than using the diffusion

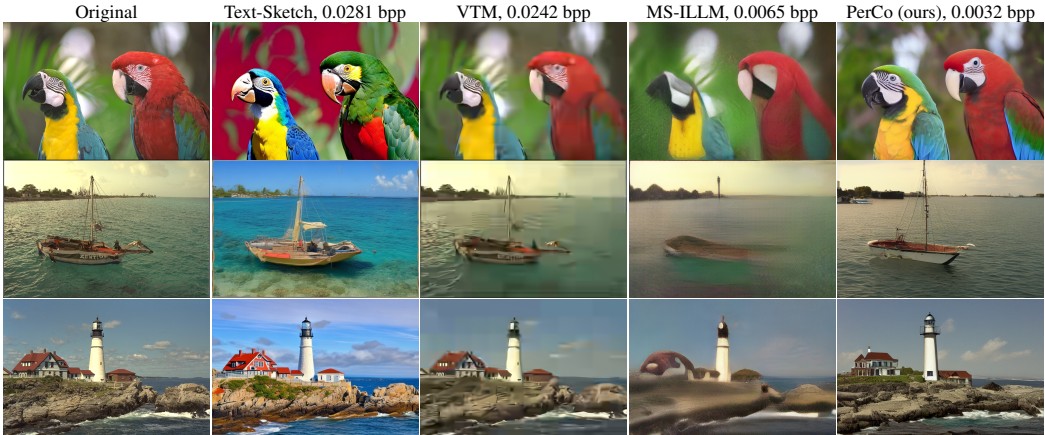

Figure 1: Kodak images compressed with the diffusion-based Text-Sketch approach PICS (Lei et al., 2023), the hand-crafted codec VTM (vtm), MS-ILLM (Muckley et al., 2023) which leverages an adversarial loss, and with PerCo (ours). Taking the lowest available bitrate for each method.

model for compression directly, we use it as a decoder in a VQ-VAE-like autoencoder. The (approximate) log-likelihood loss of diffusion models offers an alternative to the use of distortions such as MSE or LPIPS used to train most neural compression models that make strong assumptions on the conditonal distribution on images given the latents which are inappropriate at low bitrates, and lead to compression artefacts. For decoding, we sample from the conditional diffusion model, which allows us to obtain a set of reconstructions that reflect the uncertainty about the original source image. We augment the quantized image representation with a global image description that captures the high-level semantic information of the image. To this end, we use an automatically generated textual image description, *e.g.* using BLIP (Li et al., 2023). Alternatively, this can take the form of a global image feature extracted from a pre-trained image backbone. We demonstrate that this high-level information is helpful for compression performance at extremely low rates. Our work is closely related to that of Lei et al. (2023), who explored text-conditioned diffusion decoders for image compression, and the only prior work that we are aware of that considers the same ultra-low bitrates as we do. In contrast, their work uses (automatically generated) binary contour maps as a spatial conditioning signal which carry little detail on colors and textures, and leads to reconstructions that are not faithful to the original. Experimentally we observe significantly improved FID and KID scores compared to competing methods. Moreover, we find that FID and KID are much more stable across bitrates than other methods, which aligns with our goal of image compression with perfect realism.

In sum, our contributions are as follows:

- We develop a novel diffusion model called PerCo for image compression that is conditioned on both a vector-quantized latent image representation and a textual image description.
- We obtain realistic reconstructions at bitrates as low as 0.003 bits per pixel, significantly improving over previous work, see Figure 1.
- We obtain state-of-the-art FID and KID performance on the MS-COCO 30k dataset; and observe no significant degradation of FID when reducing the bitrate.

## 2 RELATED WORK

**Neural image compression codecs.** Ballé et al. (2017) demonstrated that an end-to-end neural image compression codec could outperform the classical JPEG codec in terms of rate-distortion performance. Ballé et al. (2018) enhance this approach by conditioning the latent on a "hyperprior" that encodes high-level information in the image, greatly improving performance. Follow-up works improved the latent conditioning mechanism (Minnen et al., 2018; Minnen & Singh, 2020; Cheng et al., 2020; He et al., 2022a). Optimization for rate-distortion alone leads to distributional mismatch between the compressed image distribution and the natural image distribution (Blau & Michaeli, 2018;

2019). Agustsson et al. (2019) and Mentzer et al. (2020) added an adversarial loss term and demonstrated its usefulness for human perception. Similarly He et al. (2022b) builds upon ELIC (He et al., 2022a) by adding perceptual losses including adversarial ones. Follow-up works have improved the discriminator architecture (Muckley et al., 2023), decode-time realism guidance (Agustsson et al., 2023), or investigated alternative autoencoders with discrete entropy models (El-Nouby et al., 2023). Xu et al. (2023) introduce the notion of conditional perceptual quality, where perceptual quality is dependent on side information. Our work is related to this, as the global textual or visual representation on which we condition the decoder can be seen as a form of side information.

**Diffusion models.** Ho et al. (2020) improved the original diffusion probabilistic model (Sohl-Dickstein et al., 2015) that gave it impressive performance on image generation benchmarks. Dhariwal & Nichol (2021) demonstrated superior performance to GANs for image generation with classifier guidance, while Kingma et al. (2021) improved diffusion model likelihood estimation. Conditioning diffusion models on large text-encoder networks enables these models to generate realistic images from natural language prompts (Nichol et al., 2022; Saharia et al., 2022), and defining the diffusion process in an autoencoder latent space reduces the computational cost of the diffusion process. In our work we build upon large-scale pretrained text-conditioned diffusion models to benefit from the image generation capabilities embedded in them.

**Diffusion generative compressors.** Ho et al. (2020) considered compression with diffusion models and reverse channel coding and the corresponding rate-distortion performance, but reported only experiments on $32 \times 32$ CIFAR-10 images and MSE distortion. Theis et al. (2022) demonstrated that the reverse channel coding approach gave FID numbers superior to HiFiC for $64 \times 64$ ImageNet images. Saharia et al. (2022) introduced a super-resolution diffusion model that removed artefacts from JPEG images. Ghouse et al. (2023) developed a complete version of a residual diffusion codec with images of high quality for a suite of codecs. Hoogeboom et al. (2023) showed similar results using a neural autoencoder baseline codec, and using a separate diffusion model that does not require any additional bits to refine the reconstruction. Yang & Mandt (2023) developed an alternative approach, where a hyperprior-based neural encoder was trained jointly with a diffusion decoder, also showing improved performance to HiFiC. In Pan et al. (2022), they optimize a textual embedding on top of a pretrained text-to-image diffusion model. They also design a compression guidance method used at each denoising step to better reconstruct images. Lei et al. (2023) leverage text-conditioned diffusion models for image compression at very low bitrates, using prompt inversion to encode the image into text, and adding spatial detail via compressed binary contour maps. Our work is similar, but we replace computationally costly prompt inversion (Wen et al., 2023) with fast feed-forward image captioning (Li et al., 2023), and replace binary contour maps which carry little appearance information with end-to-end learned vector-quantized image features.

## 3 PERCEPTUAL COMPRESSION WITH A DIFFUSION DECODER

### 3.1 OVERALL FRAMEWORK

The foundation of most lossy compression lies in rate-distortion theory (Shannon, 1948), which quantifies the balance between the bitrate needed to transmit a signal *vs.* the distortion of the signal. Let us assume an input signal $\boldsymbol{x}$, with its corresponding quantized representation $\boldsymbol{z}$ and reconstruction $\hat{\boldsymbol{x}}(\boldsymbol{z})$. Neural compression is commonly achieved via the minimization of a training objective formulated as a linear combination of a rate and a distortion term:

$$\mathcal{L}_{\mathrm{RD}} = \mathbb{E}_{P_{\boldsymbol{x}}}[\mathbb{E}_{P_{\boldsymbol{z}|\boldsymbol{x}}} \mathcal{L}_{\mathrm{R}}(\boldsymbol{z}) + \lambda \mathcal{L}_{\mathrm{D}}(\hat{\boldsymbol{x}}(\boldsymbol{z}), \boldsymbol{x})], \tag{1}$$

where $P_{\boldsymbol{x}}$ is the data distribution and $P_{\boldsymbol{z}|\boldsymbol{x}}$ is the posterior distribution of the quantized codes. The rate term $\mathcal{L}_{\mathrm{R}}(\boldsymbol{z})$ estimates the bitrate either by means of scalar quantization and a continuous entropy model (Ballé et al., 2017), or using vector-quantization in combination with discrete entropy models (El-Nouby et al., 2023). For simplicity, in PerCo, we employ vector-quantization combined with a uniform entropy model leading to a constant rate controlled through two hyper-parameters: the number of elements that are quantized, and the codebook size. Therefore, the rate $\mathcal{L}_{\mathrm{R}}$ can be considered fixed when training our models and can be dropped from the optimization formulation. Next, we introduce a new formulation of the distortion term, $\mathcal{L}_{\mathrm{D}}$, enabling the integration of a pre-trained diffusion model, thereby achieving a higher level of perceptual quality. Our aim with this formulation is to leverage a large pre-trained diffusion model as a robust image prior, enabling realistic image reconstruction even at extremely low bitrates.

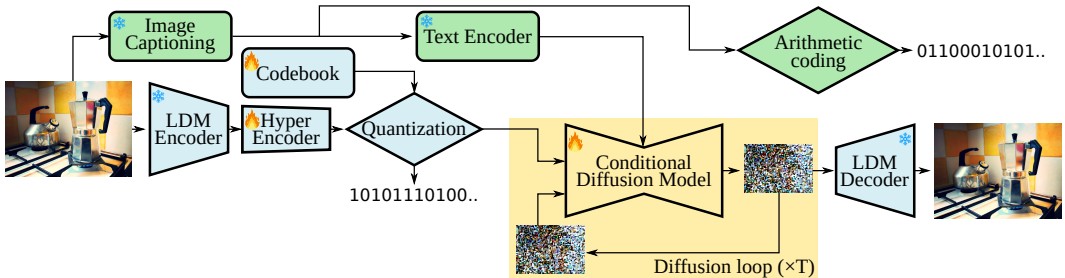

Figure 2: Overview of PerCo. The LDM encoder maps an RGB image into the latent space of the diffusion model. The "hyper encoder" then maps the image to a hyper-latent with smaller spatial resolution, which is then vector-quantized and represented as a bitstream using uniform coding. The image captioning model generates a textual description of the input image, which is losslessly compressed, and processed by a text encoder to condition the diffusion model. The diffusion model reconstructs the input image in its latent space conditioned on the output of the text encoder and the hyper encoder. Finally, the LDM decoder maps the latent reconstruction back to RGB pixel space.

Diffusion models (Sohl-Dickstein et al., 2015; Ho et al., 2020) approximate a data distribution by gradually denoising a unit Gaussian random variable. This is achieved by training a denoising function to reverse a diffusion process that takes an initial sample, denoted as $\boldsymbol{x}_0$, from the data distribution and adds small Gaussian noise over a large number of steps. In practice, a noise estimator $\epsilon_\theta(\boldsymbol{x}_t, t, \boldsymbol{z})$ implemented as a neural network is trained to denoise input data points at timesteps $t \in [0, T]$, where $\boldsymbol{x}_t = \sqrt{\alpha_t}\boldsymbol{x}_0 + \sqrt{1 - \alpha_t}\epsilon$ is a corrupted version of the original data point $\boldsymbol{x}_0$ with Gaussian noise $\epsilon$. The scalar $\alpha_t$ fixes the noise strength, varying from $\alpha_0 \simeq 1$ for no noise to $\alpha_T \simeq 0$ for pure Gaussian noise. Finally, $\boldsymbol{z}$ is an optional condition, such as a text-prompt in text-to-image models. In our case, $\boldsymbol{z}$ is the quantized representation of the input image. Once the noise estimator is trained, the model can generate samples by drawing $\boldsymbol{x}_T \sim \mathcal{N}(0, I)$, and applying iterative denoising for $t = T, \ldots, 1$. To reduce the computational cost, we use a latent diffusion model (LDM) where the variables $\boldsymbol{x}_t$ are defined in the latent space of an autoencoder.

We formulate our distortion loss within the probabilistic diffusion framework. For every the diffusion step $t$, we compute an estimation $\hat{\boldsymbol{x}}_{t-1}$ of $\boldsymbol{x}_{t-1}$ from $\boldsymbol{z}$ and $\boldsymbol{x}_t$ and minimize its error:

$$\mathcal{L}_{\text{Diff}}^t = \mathbb{E}_{P_{\boldsymbol{x}}} \mathbb{E}_{P_{\boldsymbol{z}, \boldsymbol{x}_t | \boldsymbol{x}}} \left\| \boldsymbol{x}_{t-1} - \hat{\boldsymbol{x}}_{t-1}(\boldsymbol{x}_t, \boldsymbol{z}) \right\|_2^2. \tag{2}$$

Up to a multiplicative constant, this loss can be rewritten as

$$\mathcal{L}_{\text{Diff}}^t \propto \mathbb{E}_{P_{\boldsymbol{x}}} \mathbb{E}_{P_{\boldsymbol{z}, \boldsymbol{x}_t | \boldsymbol{x}}} \mathbb{E}_{\epsilon \sim \mathcal{N}(0,1)} \left\| \epsilon - \epsilon_\theta(\boldsymbol{x}_t, \boldsymbol{z}, t) \right\|_2^2. \tag{3}$$

We obtain the final loss by taking the expectation of Eq. (3) w.r.t. the steps $t$, and use $\boldsymbol{v}$ prediction objective of Salimans & Ho (2022), which is more stable during training. The loss can be augmented with any image-level loss such as LPIPS (Zhang et al., 2018) to improve perceptual image similarity.

## 3.2 ENCODING LOCAL AND GLOBAL CONTEXT

We encode the input image as $\boldsymbol{z} = (\boldsymbol{z}_l, \boldsymbol{z}_g)$, where $\boldsymbol{z}_l$ and $\boldsymbol{z}_g$ carry complementary local and global context, respectively. The overall structure of our compression scheme is illustrated in Figure 2.

**Local spatial encoding.** We encode the input image into a quantized tensor proceeding in three stages. First, we employ the LDM encoder $E(\cdot)$ to achieve a first dimension reduction. Assuming an input image of resolution $512 \times 512$, we obtain a feature map of dimension $4 \times 64 \times 64$. Second, we add a lightweight "hyper encoder" $H$ composed of several convolutional layers to project the LDM features to a lower spatial resolution tensor $\boldsymbol{H}_s$. The spatial resolution, $h \times w$, of $\boldsymbol{H}_s$ is chosen according to the target bitrate. Third, we proceed to quantization of $\boldsymbol{H}_s$ to obtain $\boldsymbol{z}_l$ via vector-quantization. As in VQ-VAE (van den Oord et al., 2017; Razavi et al., 2019), each of the $h \times w$ vectors $\boldsymbol{h}_s$ in $\boldsymbol{H}_s$ is mapped to an element in a codebook learned using the vector quantization loss:

$$\mathcal{L}_{\text{VQ}} = \mathbb{E}_{\boldsymbol{h}_s} \left[ \| sg(\boldsymbol{h}_s) - \boldsymbol{z}_q \|_2^2 + \| sg(\boldsymbol{z}_q) - \boldsymbol{h}_s \|_2^2 \right], \tag{4}$$

where $sg(.)$ is the stop-gradient operation, and $z_q$ is the mapping of $h_s$ to its closest codebook entry. For better codebook usage, we follow Yu et al. (2022) and set the output of the hyper encoder to be relatively low dimensional: 32 in our case. We also found it beneficial to replace the codebook loss by exponential moving average on the codebooks and thus the first term in Eq. (4) is no longer used. The quantized encoder output is then upsampled to $64 \times 64$ resolution, to be channel-wise concatenated to the input of the noise estimation network of the diffusion model. The hyper-network is trained end-to-end while finetuning the diffusion model. For simplicity, we opt for uniform coding to map the quantized latents to a bitstream. Using $\log_2 V$ bits to encode each element in the $h \times w$ sized hyper latent encoding, where $V$ is the codebook size, results in a total of $hw \log_2 V$ bits.

**Global encoding with image captioning.** While $z_l$ can accurately encode local information, our experiments show that relying on $z_l$ alone leads to unsatisfying realism. Therefore, we add a global encoding of the image $z_g$ which provides additional context. For this purpose, we use an off-the-shelf state-of-the-art image captioning model, such as BLIP-2 (Li et al., 2023) or IDEFICS (Laurançon et al., 2023), which we keep fixed when training our model. Note that user-generated captions can also be used, as explored in our experiments. Similar to Lei et al. (2023), we losslessly compress the caption using Lempel-Ziv coding as implemented in the zlib library (zli) to obtain $z_g$. Alternatively, we also explore the use of global image features extracted using an image backbone network.

**Decoding with a diffusion model.** The quantized local image features in $z_l$ are fed to the denoising U-Net of the diffusion model via concatenation with the latent features $x_t$ of the current time step. To adapt to this change of dimension, we extend the kernel of the first convolutional layer of the U-Net by adding a number of channels that corresponds to the channel dimension of $z_l$ which is randomly initialized. The global encoding $z_g$ is losslessly decoded and passed to the diffusion model through cross-attention layers of the pre-trained diffusion model.

Furthermore, we found it beneficial to use classifier-free guidance (Ho & Salimans, 2021), which we apply at inference time for the text conditioning $z_g$, *i.e.* we contrast conditioning on local features $z_l$ alone *vs.* conditioning on both $z_l$ and $z_g$. This leads to the noise estimate:

$$\hat{\epsilon}_\theta = \epsilon_\theta(x_t, (z_l, \emptyset), t) + \lambda_s \left( \epsilon_\theta(x_t, (z_l, z_g), t) - \epsilon_\theta(x_t, (z_l, \emptyset), t) \right), \tag{5}$$

where $\lambda_s$ is the guidance scale. We empirically found $\lambda_s = 3$ to work well. To enable classifier-free guidance, we train PerCo by dropping the text-conditioning in $10\%$ of the training iterations. When dropping the text-conditioning we use a constant learned text-embedding instead.

## 4 Experiments

### 4.1 Experimental Setup

**Implementation details.** We base our model off a text-conditioned latent diffusion model, with latent space provided by a convolutional autoencoder and diffusion model with an architecture similar to GLIDE (Nichol et al., 2022). The model is pre-trained on a proprietary dataset consisting of around 300M image-caption pairs. We also experiment with an image-conditioned model. We use BLIP-2 (Li et al., 2023) to obtain image captions for conditioning the decoder, and limit the maximum caption length to 32 tokens. The hyper encoder consists of nine residual convolutional blocks, and includes a number of downsampling layers depending on the spatial map resolution selected for the target bitrate. In total, the hyper encoder contains between 4M and 8M parameters. During all our experiments, the autoencoder weights are frozen. We train the hyper encoder and finetune the diffusion model on OpenImages (Kuznetsova et al., 2020), similar to Lei et al. (2023) and Muckley et al. (2023). We use random $512 \times 512$ crops, and instead of finetuning the full U-Net we found it beneficial to only finetune the linear layers present of the diffusion model, representing around 15% of all weights.

**Datasets.** For evaluation, we use the **Kodak** dataset (Franzen, 1999) as well as **MS-COCO 30k**. On COCO we evaluate at resolution $256 \times 256$ by selecting the same images from the 2014 validation set (Lin et al., 2014) as Hoogeboom et al. (2023) and Agustsson et al. (2023). We evaluate at resolution $512 \times 512$ on the 2017 training set (Caesar et al., 2018), which is the same resolution used for evaluation by Lei et al. (2023), and use captions and label maps for some metrics.

**Metrics.** To quantify image quality we use **FID** (Heusel et al., 2017) and **KID** (Bińkowski et al., 2018), which match feature distributions between sets of original images and their reconstructions.

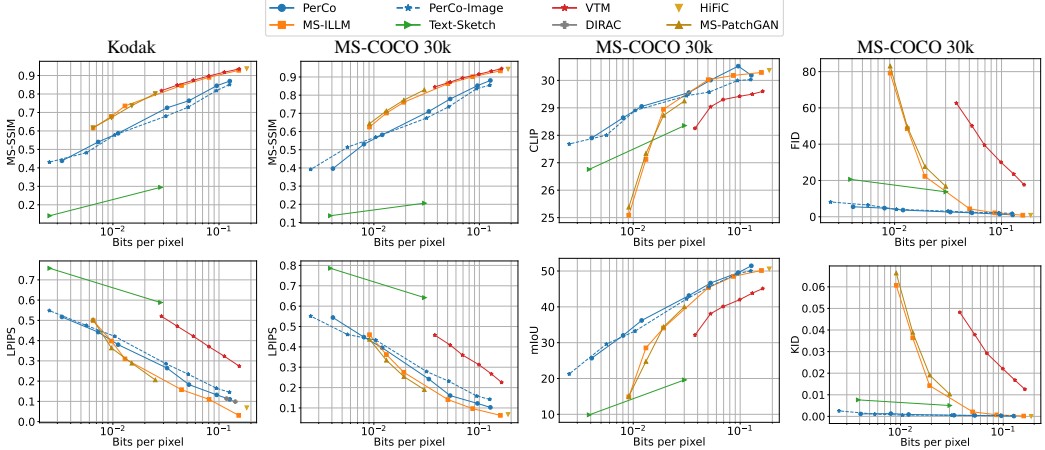

Figure 3: Evaluation of PerCo and other image compression codecs on Kodak and MS-COCO 30k.

We measure distortion using **MS-SSIM** (Wang et al., 2003) and **LPIPS** (Zhang et al., 2018). In App. B.1 we also report **PSNR**, but find these low-level distortion metrics to be less meaningful for low rates, see Fig. 14. Therefore we include other more semantic-level metrics. We compute the **CLIP score** to measure global alignment of reconstructed samples with ground truth captions (Hessel et al., 2021). Unless otherwise specified, we condition our model on captions generated by BLIP2, which are different from the ground-truth captions used to compute the CLIP score. Second, we adopt the mean Intersection over Union (**mIoU**) metric to measure the local semantic preservation, see *e.g.* Sushko et al. (2022). In this case, we pass reconstructed images through a pre-trained semantic segmentation model, and compare the predicted semantic maps to the ground truth ones. We take the pretrained ViT-Adapter segmentation network (Chen et al., 2023), which was trained on the COCO train split. For Kodak we only report LPIPS and MS-SSIM, as the dataset is too small (24 images) to reliably compute FID and KID, and does not come with semantic annotations.

**Baselines.** We include representative codecs from several families. VTM is a state-of-the-art handcrafted codec used in the VVC video codec (vtm). Among neural compressors using adversarial losses, we include MS-ILLM (Muckley et al., 2023) which improves over HiFiC (Mentzer et al., 2020) using an adversarial loss based on multi-class classification. For HiFiC we include their model with the lowest available bitrate, and add MS-PatchGAN, which is a close reproduction of HiFiC by Muckley et al. (2023). We compare to multi-realism approach of Agustsson et al. (2023), at their lowest bitrate with the best realism. Text-Sketch (Lei et al., 2023) is the only prior work we are aware of that evaluates image compression at bitrates below 0.01, it is based on text-conditioned diffusion models. We include two other recent diffusion-based approaches: DIRAC (Ghouse et al., 2023) and HFD/DDPM (Hoogeboom et al., 2023).

## 4.2 MAIN RESULTS

**Comparison to state-of-the-art methods.** In Fig. 3 we compare our results to state-of-the-art codecs. We observe that PerCo (ours) yields significantly lower (better) FID and KID compared to other approaches at lower bitrates (<0.04 bpp), and that our FID and KID curves are much flatter, indicating a decoupling of realism and bitrate that is not seen for other methods. For the semantics-related CLIP and mIoU metrics we also obtain consistent improvements over all other methods, in particular at low rates. For LPIPS, PerCo is better than Text-Sketch and VTM, but somewhat worse than other approaches for rates >0.01 bpp. Similarly, for MS-SSIM we improve over Text-Sketch, but are worse than other

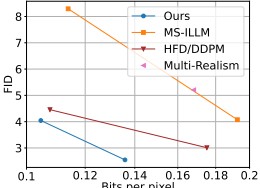

Figure 4: Evaluation on COCO at $256 \times 256$ res.

methods. However, as we illustrate in Figs. 10 and 14 in App. B.2, such similarity metrics are not necessarily meaningful at low bitrates, as they do not capture realism.

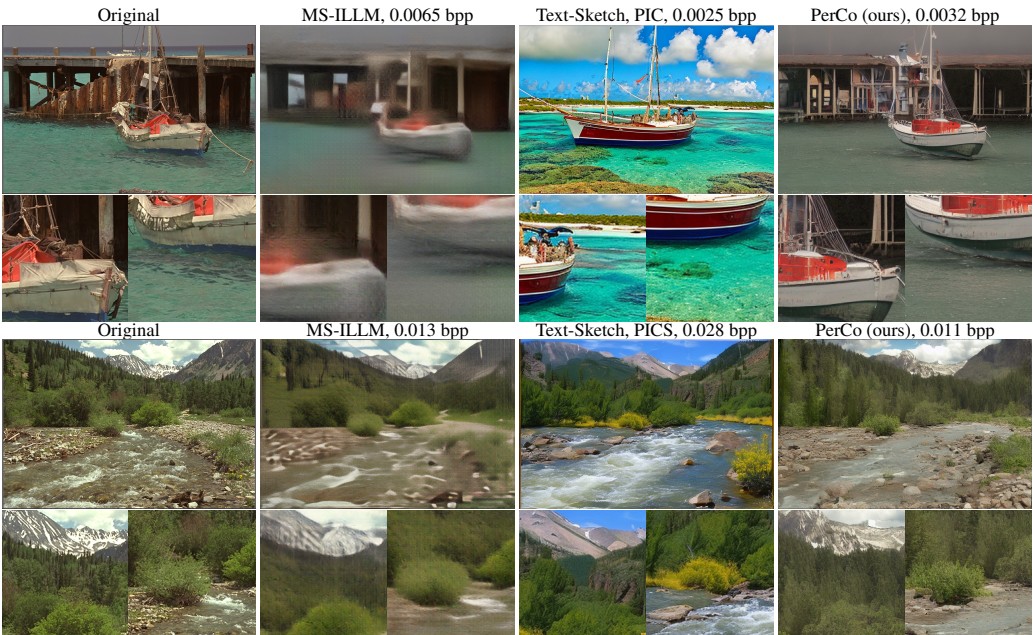

Figure 5: Comparing PerCo on images from the Kodak dataset to MS-ILLM (Muckley et al., 2023) which leverages an adversarial loss, and the diffusion-based Text-sketch approach (Lei et al., 2023) conditioned on a text only (top, PIC, 0.0025 bpp) and text + sketch (bottom, PICS, 0.028 bpp).

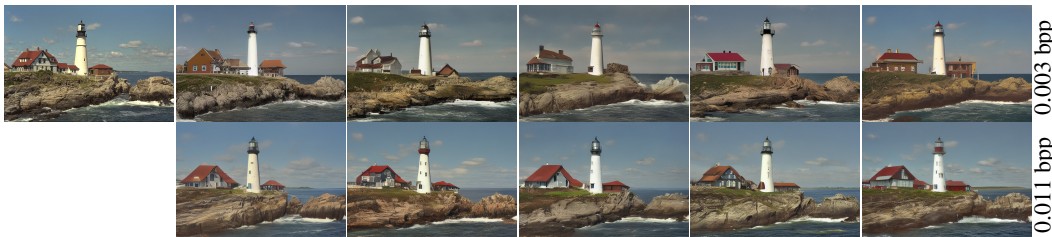

Figure 6: Reconstruction sampled of a Kodak image at 0.003 and 0.011 bpp. Best viewed zoomed.

In Fig. 4 we compare to HFD/DDPM (using their lowest bitrates: 0.107 and 0.175 bpp) on FID on MS-COCO 30k at $256 \times 256$ resolution, as used in their paper. We also include MS-ILLM and the multi-realism approach of Agustsson et al. (2023). We find that PerCo yields best FID for both rates.

**Qualitative comparisons.** We provide qualitative comparisons of PerCo to baselines in Fig. 1 and Fig. 5. PerCo yields reconstructions of higher quality with less artefacts, more faithful colors, and realistic details. VTM and MS-ILLM reconstructions are blurry and/or blocky, while Text-Sketch strongly deviates from the originals in color. When using text-only conditioning for Text-Sketch (PIC, 0.0025 bpp), it is not able to achieve spatial alignment, by lack of local image encoding.

### 4.3 ABLATIONS

**Diversity in reconstructions.** With our model, reconstructions are sampled from a conditional diffusion model. Ideally, the conditional distribution concentrates all mass perfectly on the original input image. In practice, this does not happen due to limited model capacity, training data, and bitrate. In particular, we expect that at lower bitrates —where there is less information about the original sample— the conditional distribution would have more variance, and we expect to observe more diversity among the samples. The reconstruction at two bitrates in Fig. 6 confirm this.

**Conditioning modalities.** In Fig. 7, we evaluate PerCo using each conditioning modality separately (textual or spatial), to analyze the contribution of each. We compare BLIP-2 (Li et al., 2023) to generate captions (default) to using IDEFICS (Laurançon et al., 2023) which produces more detailed descriptions, and ground-truth (GT) captions. When not using descriptive captions, we consider

Figure 7: Ablation of conditioning on local features and/or different captions. See text for detail.

Table 1: Ablations of classifier-free guidance forms (left), and impact of conditional diffusion model and quantization on reconstruction abilities (right). All models at 0.0112 bpp, except for LDM autoencoder and PerCo w/o quantization which use single precision for non-quantized latents.

| Guidance type | ↓FID | ↓LPIPS | ↑SSIM | ↑mIoU | ↑CLIP |
|---|---|---|---|---|---|
| None | 7.89 | 0.40 | 0.58 | 30.84 | 28.36 |
| Text and spatial | 13.84 | 0.41 | 0.58 | 32.26 | 28.87 |
| PerCo: Text only | 4.42 | 0.39 | 0.58 | 46.64 | 29.05 |

| Method | ↓FID | ↓LPIPS | ↑SSIM | ↑mIoU | ↑CLIP |
|---|---|---|---|---|---|
| LDM autoencoder | 0.67 | 0.07 | 0.92 | 51.04 | 30.19 |
| PerCo w/o quant. | 0.68 | 0.07 | 0.92 | 51.03 | 30.18 |
| PerCo (0.012 bpp) | 4.42 | 0.39 | 0.58 | 46.64 | 29.05 |

conditioning on a learned textual embedding that is constant across images. We also tried using an empty text or on a fixed generic text — *"A high quality photograph."* — for all images, and observed similar or slightly worse performance as the learned text embedding setting.

The model using the constant text embedding (PerCo w/ only Spatial Encoding) performs significantly worse than the other models in terms of FID and CLIP score. For the LPIPS and mIoU metrics, which rely on the alignment of the original and reconstructed images, the results are comparable to other models. This can be attributed to the fact that accurate alignment is primarily achieved through local feature conditioning, which remains unchanged in this model. Consistently, we also notice that when using textual condition only (blue, orange, and red square), the performance in terms of LPIPS and mIoU is worse than for the models that use spatial conditioning, while the CLIP scores are in line with the models using both textual and spatial conditioning. Both types of conditioning contribute to improving FID, and the text-only conditioned models perform in between the models with only spatial conditioning and those with both types of conditioning.

Between models using captions from BLIP (blue curves) and IDEFICS (orange curves) in combination with spatial conditioning, we observe similar FID scores and somewhat better CLIP scores for the more detailed IDEFICS captions. In terms of LPIPS and mIoU we find that the models with BLIP captions perform better, which is largely due to bitrate required to encode the longer captions: leaving out the textual encoding cost, both models perform very similarly. Overall, using ground-truth captions (GT, red curves) yields similar results as using BLIP, while BLIP and IDEFICS yield slightly better mIoU and CLIP scores. As COCO contains five GT captions per image, we ensure that when conditioning on a GT caption, a different one is used to compute the CLIP score.

**Classifier-free guidance.** In Tab. 1 (left), we compare three options regarding classifier-free guidance (CFG) in our model (i) not using CFG, (ii) applying CFG on the text and spatial feature maps, *i.e.* contrasting full conditioning to no conditioning (we use an constant learned text embedding, and fixed all-zeros feature maps for local conditioning), and (iii) applying CFG to text-conditioning only, *i.e.* always conditioning on the spatial feature map. We find that best results are obtained when CFG is applied to textual conditioning only, which yields significantly better FID and mIoU metrics.

**Impact of autoencoder and quantization.** To assess to which extent performance is bounded by the LDM autoencoder or the use of limited bitrate, we compare PerCo to the LDM encoder/decoder reconstruction and to PerCo trained without quantization in Tab. 1 (right). We observe that the LDM autoencoder significantly improves all metrics, and that PerCo trained without quantization obtains roughly the same performance as the autoencoder. This shows that PerCo is a strong conditional model, and that its performance is mainly bounded by the quantization module which throttles the bitrate of the image encoding, but could also lead to suboptimal training through noisy gradients.

Figure 8: Evolution of FID, LPIPS, MS-SSIM and CLIP scores depending on the number of denoising steps for different bitrates. In all cases the model has been trained using 50 denoising steps.

**Number of timesteps as a tradeoff between perception and distortion.** In Figure 8, we show how FID, LPIPS, MS-SSIM and CLIP scores vary when we change the number of denoising steps. We do the evaluation for seven different bitrates, ranging from 0.0032 to 0.1263 bpp. First, we notice that for higher bitrates, the metrics are quite constant across the number of timesteps, which means that the model has less uncertainty about the image to reconstruct and obtains optimal performance after only few denoising steps. Secondly, we observe a tradeoff between distortion and realism for lower bitrates. Indeed, when the number of timesteps is small, FID is worse, but MS-SSIM is optimal, while this phenomenon reverses when increasing the number of denoising steps.

**Conditioning on global image features.** Besides the text-conditioned model described above and used in most experiments (PerCo), we also experimented with a model conditioned on image embeddings extracted from an image backbone network (PerCo-Image). While we use lossless Lempel-Ziv coding to compress the captions for the text-conditioned model, we use product quantization (Jégou et al., 2011) for lossy compression of the image embeddings. We divide the embeddings into $M$ subvectors and quantize each subvector using a separate codebook with $V$ elements. We empirically found that we achieve similar performance compared to training with non-quantized image embeddings with $M = 16$ and $V = 1024$. For $512 \times 512$ images, the bitrate obtained for the image embeddings is 0.00061 bpp, which is $3.5 \times$ lower than the bitrate of 0.00219 bpp for BLIP image captions. From the results in Fig. 3, we observe that the model with global image embeddings obtains competitive results, but has slightly worse pairwise image reconstruction metrics at high bitrates in terms of LPIPS and MS-SSIM on Kodak and MS-COCO 30k.

## 5 CONCLUSION

We proposed PerCo, an image compression model that combines a VQ-VAE-like encoder with a diffusion-based decoder, and includes an second conditioning stream based on textual image descriptions. The iterative diffusion decoder allows for more realistic reconstructions, in particular at very low bitrates, as compared to feed-forward decoders used in previous work, even when trained with perceptual and adversarial losses. With our work we make step towards perfect realism codecs: we observe realistic reconstructions at bitrates as low as 0.003 bits per pixel, a bitrate regime that to our knowledge is explored in only one prior work (Lei et al., 2023). We find that semantics-driven metrics such as CLIP score and mIoU are improved overall and in particular for low rates, and FID and KID are dramatically improved at low bitrates and much more stable across different bitrates.

**Limitations.** In this study we focused on medium sized images up to $768 \times 512$ resolution, similar to prior work on diffusion-based methods (Theis et al., 2022; Lei et al., 2023) that evaluated on $64 \times 64$ and $512 \times 512$ images, respectively. Extension to higher resolutions can possibly be achieved using a patch-based approach, see *e.g.* Hoogeboom et al. (2023). PerCo exhibits somewhat poorer reconstruction performance in terms MS-SSIM, PSNR and LPIPS (the latter for rates >0.01), than existing approaches. This seems at least in part to be due to limitations of the LDM autoencoder, see App. B.2. It is also probably explained by the tradeoff existing between distortion and perception (Blau & Michaeli, 2019). We leave detailed study of these two points to future work.

ETHICS STATEMENT

The use of machine learning models, in general and for image compression in particular, can lead to biases in the model related to the training data, and potentially work better on data similar to those used to train the models. Therefore the models as described in this paper should be used for research purposes only, and not deployed in practice without extensive analysis of their potential biases.

ACKNOWLEDGEMENTS

The photo in Figure 2 is by Emily Allen, licensed as CC BY 2.0, and taken from `https://www.flickr.com/photos/99453901@N00/20263907283`. We would like to thank Oron Ashual, Uriel Singer, Adam Polyak and Shelly Sheynin for sharing the text-to-image model.

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

## A  EXPERIMENTAL DETAILS

**Evaluation of third-party models.**  Where possible we reported evaluation metrics reported in the original papers. For Text-Sketch Lei et al. (2023), we used the publicly released checkpoints, and used them for evaluation with the provided code. Note that Text-Sketch resized all images at resolution 512 for evaluation on the DIV2K and CLIC 2021 datasets, which makes the comparison with other baselines not immediate. We prefer to evaluate on original images instead, and for that purpose used the Kodak dataset which is commonly used to evaluate image compression models, and contains images of $512 \times 768$ pixels. We also evaluate on the MS-COCO 2014 validation set, which is less commonly used, but which offers ground truth semantic segmentation maps and descriptive captions which enable evaluation with the semantics oriented mIoU and CLIP score metrics. For MS-ILLM and MS-PatchGAN (Muckley et al., 2023), checkpoints for models trained at additional low bitrates were obtained from the authors through personal communication. For HFD/DDPM (Hoogeboom et al., 2023), DIRAC (Ghouse et al., 2023) and Multi-Realism (Agustsson et al., 2023), we obtained the numbers shown in their paper through personal communication, as there are no available github. To evaluate VTM, we followed same the procedure as described by Muckley et al. (2023) in appendix B of their paper. For HifiC (Mentzer et al., 2020), we took the checkpoints available from their github.[1]

**Additional implementation details.**  We take a pretrained latent diffusion model doing text-to-image generation that contains 1.4B parameters. We fix the weights of the LDM autoencoder and finetune 15% of the diffusion model weights. We train our hyper-encoder for 5 epochs with AdamW optimizer at a peak learning rate of 1e-4 and a weight decay of 0.01, and a batch size of 160. We apply linear warmup for the first 10k training iterations. For bitrates higher that 0.05, we found it beneficial to add a LPIPS loss in image space to reconstruct more faithfully images. As this loss requires to backpropagate gradients through the LDM decoder (even though the LDM decoder is frozen), it is more memory intensive and we had to decrease the batch size to 40. For the vector quantization module, we used the vector-quantize-pytorch library.[2]

To finetune our latent diffusion model we used a grid of 50 timesteps. At inference time, we use 20 denoising steps to obtain good performance for the lowest bitrates. For bitrates of 0.05 and higher, we use five denoising steps as this is enough to obtain optimal performance, see Fig. 8 from the main paper. Besides, we sample images with DDIM deterministic sampling schedule, *i.e.* by fixing $\sigma_t = 0$ for all $t \in [0, T]$, where $T$ is the number of denoising steps, see Song et al. (2021).

When testing different textual conditioning modalities (either BLIP or IDEFICS) in our ablation study in Fig. 7, the conditioning used at inference time is also used when training the models. When using human-specified captions, we use PerCo trained on BLIP captions, as there are no ground-truth captions for the OpenImagesv6 dataset we used to train our model.

We use the clean-fid library to compute FID and KID.[3] For CLIP score, we compute image and text embeddings with the CLIP backbone ViT-B/32. LPIPS is calculated using the original github[4], and MS-SSIM from the FAIR Neural Compression library.[5]

**Details on the bitrates used to evaluate our models.**  To achieve a specific target bitrate, we take into account the average bitrates required to encode the textual conditioning, and then set the spatial grid size and vocabulary size of the local conditioning accordingly. More specifically, for COCO at resolution $512 \times 512$, the average textual bpp for BLIP is 0.00219, while it is 0.00622 for IDEFICS, and 0.00271 for ground truth captions. For Kodak at resolution $768 \times 512$, it is 0.00141 with BLIP and 0.00415 bpp with IDEFICS. In the experiments, we select the spatial size and codebook dimension of the quantization module to adjust the spatial encoding bitrate depending on the targeted total bitrate. We found that codebooks with less than 64 elements were detrimental to the training, so we only used larger codebooks. Table 3 lists the bpp for the spatial coding for the configurations we used to proeuce the results in our experiments. For the ablation study in Fig. 7 we used other

---

[1] https://github.com/tensorflow/compression/tree/master/models/hific
[2] https://github.com/lucidrains/vector-quantize-pytorch
[3] https://github.com/GaParmar/clean-fid
[4] https://github.com/richzhang/PerceptualSimilarity
[5] https://github.com/facebookresearch/NeuralCompression

combinations of spatial grid size and vocabulary size for the tested variants to approximately match the bitrates of our default PerCo model.

**Inference Speed.** We run benchmarking on A100 gpus using 20 denoising steps, and 5 denoising steps for bitrates higher than 0.05 bits per pixel. We compute the runtime for all the Kodak images at resolution 512x768 and provide the average time in seconds with its standard deviation. The encoding complexity of PerCo is quite similar to adversarial-based compression methods such as MS-ILLM and HifiC (which has the same architecture as MS-ILLM). Encoding with Text-Sketch (PIC/PICS) is orders of magnitude slower because it is optimizing a text embedding for each image. The decoding complexity of PerCo using 20 denoising steps (bitrates below 0.05 bpp) is comparable to that of Text-Sketch (PIC) and about 8 times faster than Text-Sketch (PICS), because PICS is conditioning on sketches which takes time to decompress. When using only five denoising steps, the decoding complexity drops rouhgly by a factor four, showing that the decoding is dominated by evaluations of the denoising network. Compared to HiFiC and MS-ILLM, the PerCo decoder using 5 denoising steps is about eight times slower, and when using 20 steps this becomes a factor 32. Please note that the running time of VTM is computed on CPU, so it may explain the longer runtime. About the diffusion-based models DIRAC and HFD/DDPM, we couldn't evaluate their models as there is no available codebase, but we found some evaluations in DIRAC paper about decoding time. They report a decoding time which is about four times slower than what they report for HifiC, which also shows that diffusion-based methods are slower than adversarial-based ones.

Table 2: Encoding and Decoding Speed (in seconds).

| Model | Encoding Speed (in sec.) | Decoding Speed (in sec.) |
|---|---|---|
| VTM | $16.892 \pm 7.574$ | $0.135 \pm 0.002$ |
| MS-ILLM | $0.084 \pm 0.021$ | $0.080 \pm 0.007$ |
| Text-Sketch (PIC) | $163.070 \pm 0.380$ | $2.725 \pm 0.012$ |
| Text-Sketch (PICS) | $190.231 \pm 2.476$ | $19.288 \pm 0.251$ |
| PerCo - 5 denoising steps | $0.080 \pm 0.018$ | $0.665 \pm 0.009$ |
| PerCo - 20 denoising steps | $0.080 \pm 0.018$ | $2.551 \pm 0.018$ |

The average decoding times for the 24 images of Kodak dataset are 0.67 secs. and 2.54 secs. when decoding with 5 and 20 steps, respectively. Timings is performed when decoding one image at a time.

Table 3: Summary of the different combinations of spatial grid size and codebook size used for spatial encoding of $512 \times 512$ images.

| Spatial size | Codebook size | Spatial bpp |
|---|---|---|
| $64 \times 64$ | 256 | 0.1250 |
| | 64 | 0.0937 |
| $32 \times 32$ | 8196 | 0.0507 |
| | 256 | 0.0313 |
| $16 \times 16$ | 1024 | 0.0098 |
| $8 \times 8$ | 1024 | 0.0024 |
| | 256 | 0.0019 |

## B    ADDITIONAL EXPERIMENTAL RESULTS

### B.1    QUANTITATIVE RESULTS

In Figure 9, we complete Figure 3 from the main paper with the PSNR metric computed on Kodak and COCO. We observe that PerCo yields consistently better (higher) PSNR than Text-Sketch, but also consistently worse (lower) than MS-ILLM. To understand the reason behind this relatively low

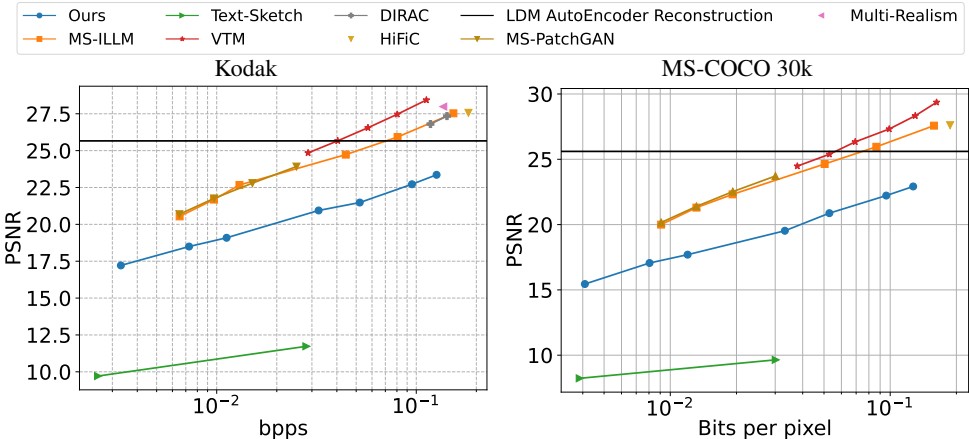

Figure 9: PSNR on Kodak and MS-COCO30k

PSNR, we also include the PSNR of our LDM autoencoder (black horizontal line), which does not involve quantization. Interestingly, we observe that the LDM autoencoder achieves a PSNR performance comparable to MS-ILLM at a bitrate of 0.08 and VTM at a bitrate of 0.04. The PSNR of the LDM autoencoder presumably explains the lower PSNR performance we observe with PerCo across different bitrates. To gain further insights into this behavior, we visually inspect the reconstruction for a Kodak image in Figure 10. From the example, the LDM autoencoder seems to preserve more relevant detail and texture than the MS-ILLM reconstruction, despite the lower PSNR.

In Figure 11, we provide an ablation on the addition of the LPIPS loss, which shows that it leads to small improvements for both LPIPS and FID at higher bitrates.

In Figure 16, we provide evaluation on an additional global image similarity metric that consists in computing CLIP scores with a cosine similarity between CLIP image embeddings of original and reconstructed images, instead of cosine similarity between text captions and reconstructed images. This is motivated by the fact that we noticed that CLIP text score was saturating at high bitrates, as it reaches the upper bound of 30.4 when computed on real images instead of reconstructed ones, while CLIP image score threshold is 1 when evaluated on real images which is higher than what is obtained at high bitrates for all the baselines.

## B.2 QUALITATIVE RESULTS

**Ablations on text.** In this ablation study, we conduct a detailed analysis to comprehend the effects of employing text as a global embedding. In Figure 12, we present qualitative examples across two bitrates, where we manipulate the conditioning text. At lower bitrates for the spatial conditioning (first and second rows), we observe that textual conditioning has a significant influence on the reconstructed image. It can even alter the semantics of the scene, such as transforming a lighthouse into a castle (second row, third column). Furthermore, we note the positive impact of text conditioning, as image quality substantially deteriorates when text is not used (first row, second column). Conversely, when the bitrate of the local encoding is higher (last row), we notice that text has minimal influence on the reconstructed image.

**Reconstruction at different bitrates.** In Figure 13, we show reconstructions of two MS-COCO 30k images using PerCo trained for different targeted spatial rates, ranging from 0.0046 to 0.1279 bpps. The image quality is quite stable across bitrates, while the similarity with the original images increases with the bitrate.

**Qualitative analysis of LDM autoencoder reconstruction** When computing the averaged similarity scores on Kodak dataset for the LDM autoencoder, we obtain 0.0775 LPIPS, 0.9131 MS-SSIM and 25.6003 PSNR which is lower than MS-ILLM at 0.1535 bpp for all the scores and lower than MS-ILLM at 0.0806 bpp on PSNR. In Figure 10 we show a Kodak image illustrating that. Even

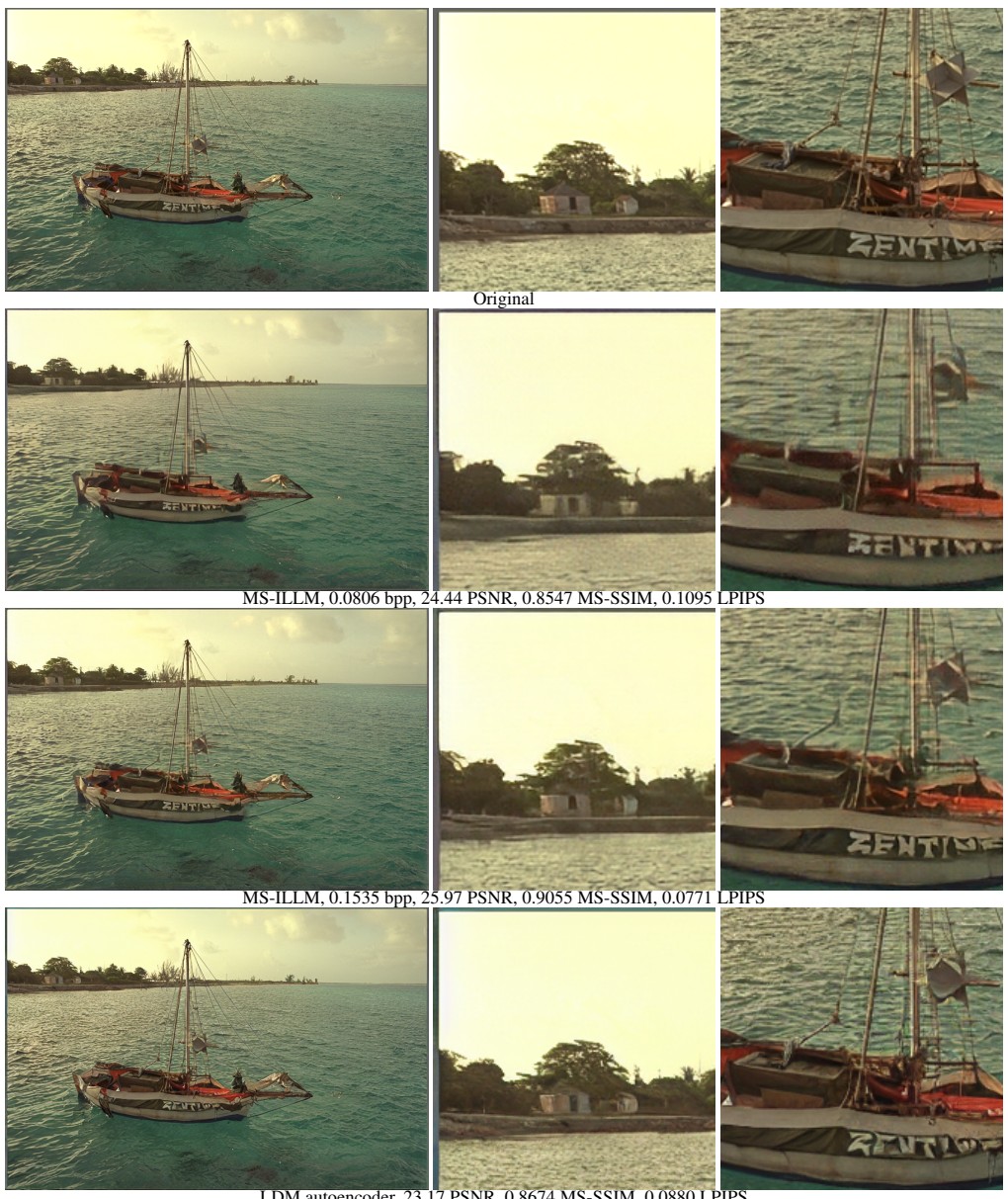

Original

MS-ILLM, 0.0806 bpp, 24.44 PSNR, 0.8547 MS-SSIM, 0.1095 LPIPS

MS-ILLM, 0.1535 bpp, 25.97 PSNR, 0.9055 MS-SSIM, 0.0771 LPIPS

LDM autoencoder, 23.17 PSNR, 0.8674 MS-SSIM, 0.0880 LPIPS

Figure 10: Qualitative comparison of reconstructions using MS-ILLM and our LDM autoencoder at comparable PSNR of a Kodak image (left) along with two zooms (middle, and right).

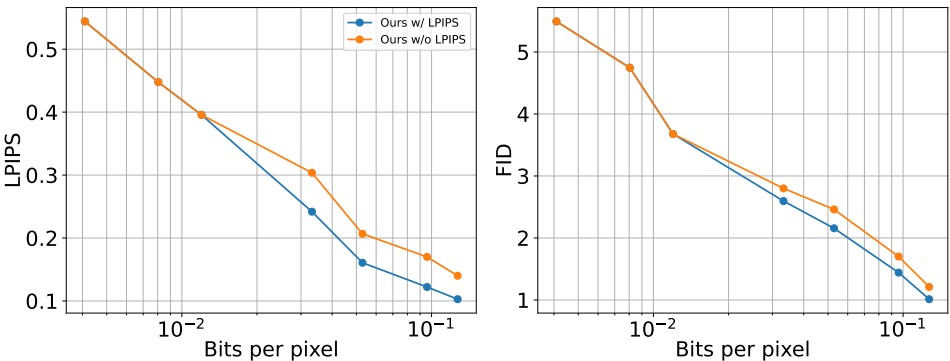

Figure 11: Ablation of adding the LPIPS loss to train the diffusion decoder.

if the LDM autoencoder produces highly detailed images, we observe that reconstruction fidelity is somewhat worse than MS-ILLM at higher bitrates. Indeed, when looking at image details such as the waves, we notice that MS-ILLM exhibits some blurring at 0.0806 bpp while LDM produces sharp details for a similar PSNR. When comparing with MS-ILLM at the higher bitrate of 0.1535 bpp, we notice that the letters on the boat are less faithful for our LDM Autoencoder than MS-ILLM with respect to the original image.

**Qualitative analysis of pixelwise similarity metrics**   We observed that pixelwise distortion metrics such as PSNR, MS-SSIM and LPIPS are not very meaningful at very low bitrates, as already pointed out by Lei et al. (2023). In the example in Figure 14, PerCo at 0.0031 bpp is 3.33 points worse in PSNR, 0.1766 points worse in MS-SSIM and 0.085 points worse in LPIPS than MS-ILLM at 0.0065 bpp. However, at these very low bit-rates PerCo yields a much sharper and realistic reconstruction than MS-ILLM, and preserves most of the semantics of the scene. These examples provide some indication that these metrics do not align well with human preference for image quality at very low bitrates.

**Additional comparisons to baselines**   In Figure 15, we compare PerCo at 0.1264 bpp, the highest bitrate we trained on, to VTM, HiFiC and MS-ILLM at similar bitrates. At this bitrate, PerCo faithfully reconstructs the original images with similarly detailed textures as HiFiC and MS-ILLM, and fewer artifacts than VTM.

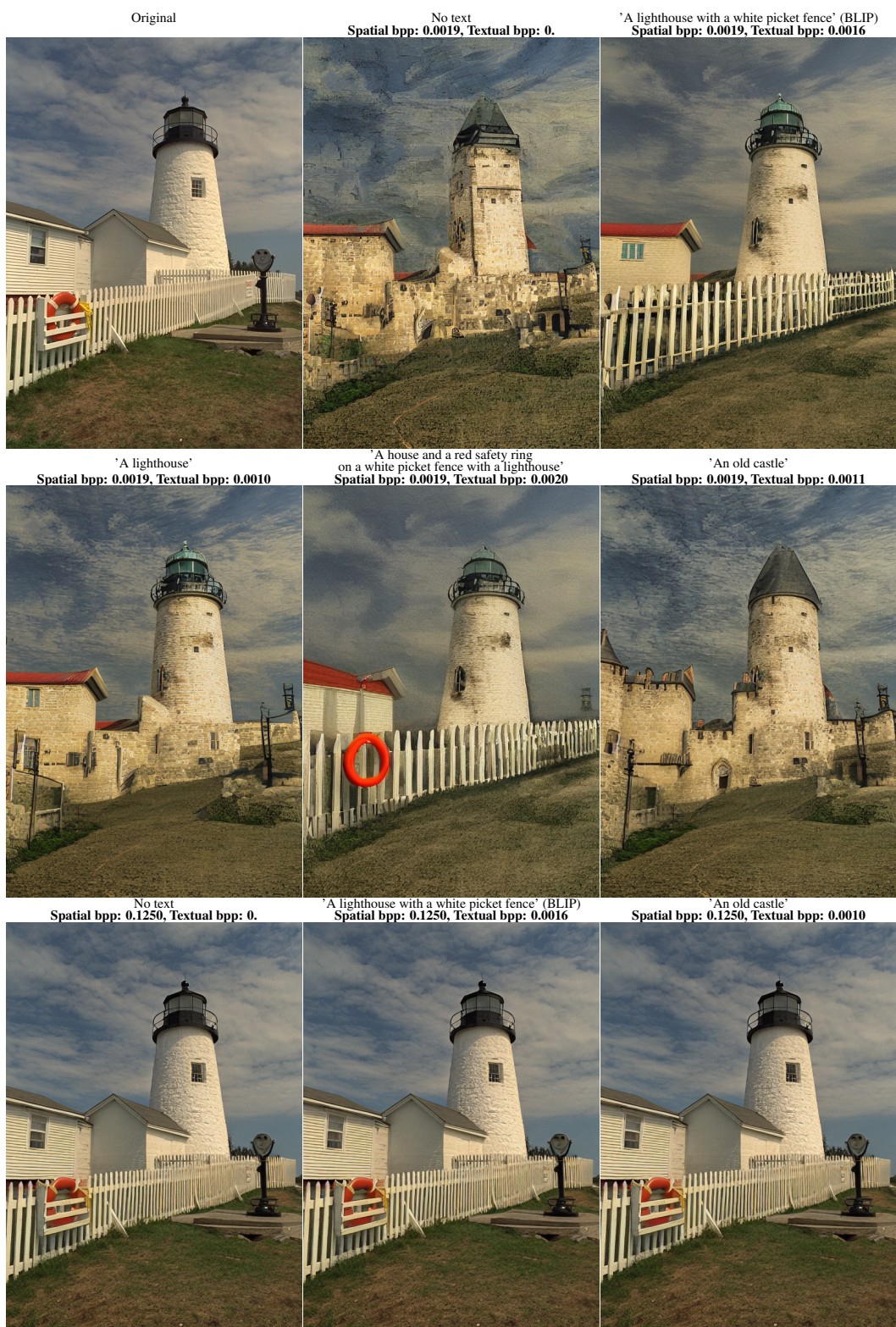

Figure 12: Reconstructions of a Kodak image when varying the global textual conditioning with a spatial bpp of 0.0019 (first two rows) and 0.1250 bpp (last row). Samples with the same spatial bpp are reconstructed from the same initial Gaussian noise from the same model.

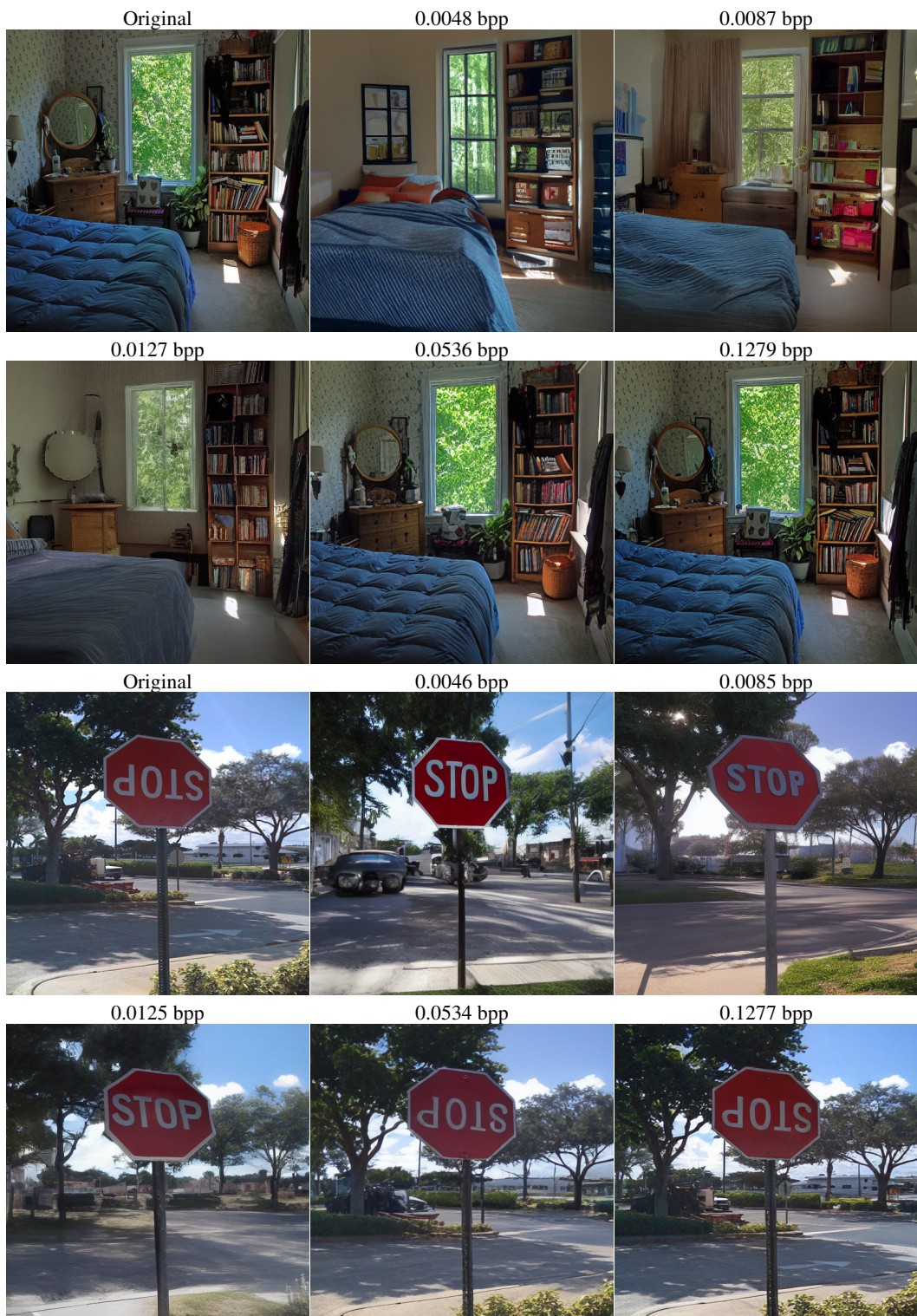

Figure 13: Reconstructions of two MS-COCO 30k images by PerCo at different bitrates. Textual conditionings are *"Bedroom scene with a bookcase, blue comforter and window"* and *"A stop sign is mounted upside-down on it's post"*, respectively.

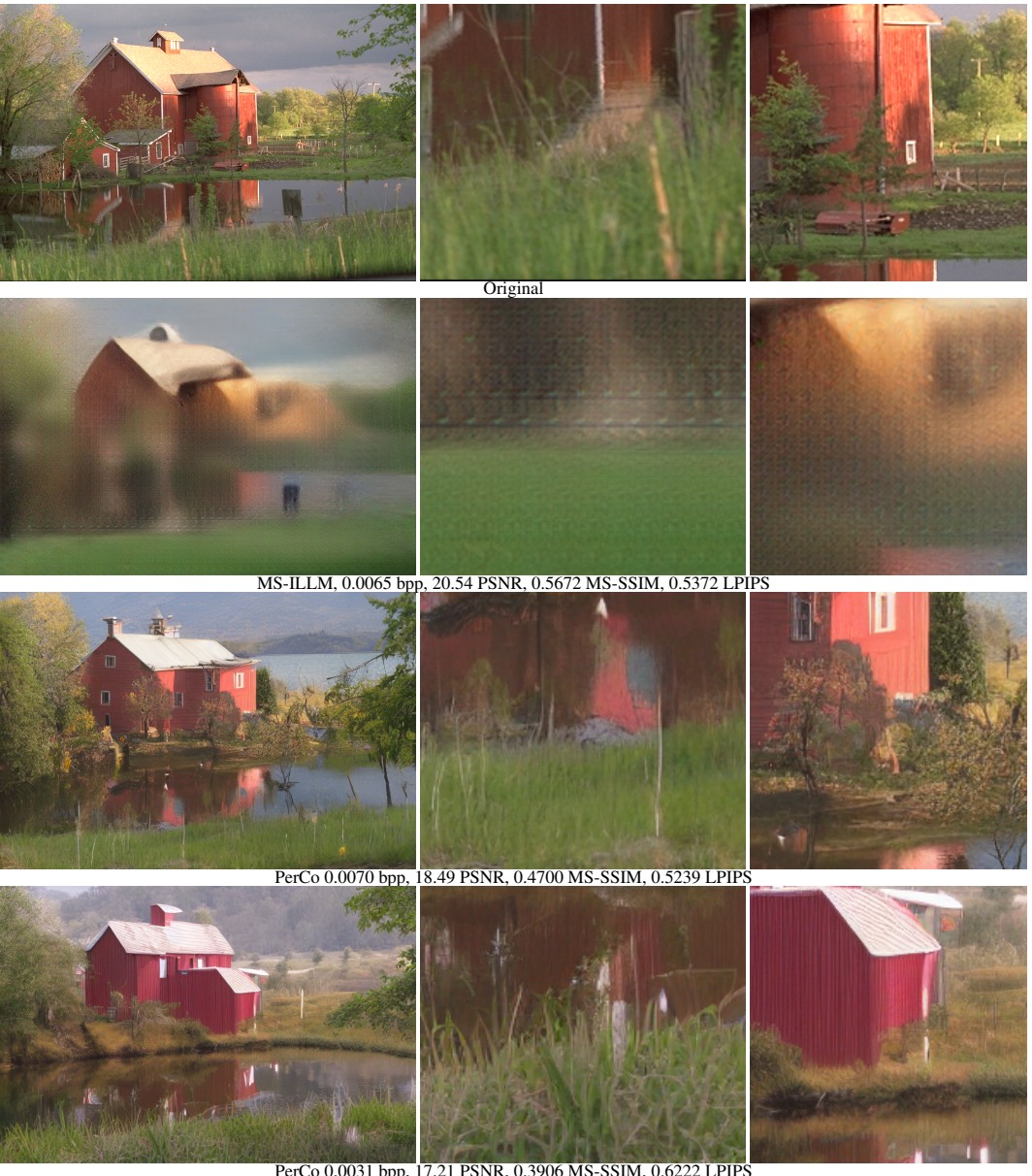

Figure 14: Qualitative comparison of reconstructions using MS-ILLM and PerCo at low bitrates of a Kodak image (left) along with two zooms (middle, and right). MS-ILLM yields a very blurry recontruction at this operating point with PerCo yields a more realistic reconstruction, yet PerCo obtains a substantially lower PSNR, MS-SSIM and LPIPS.

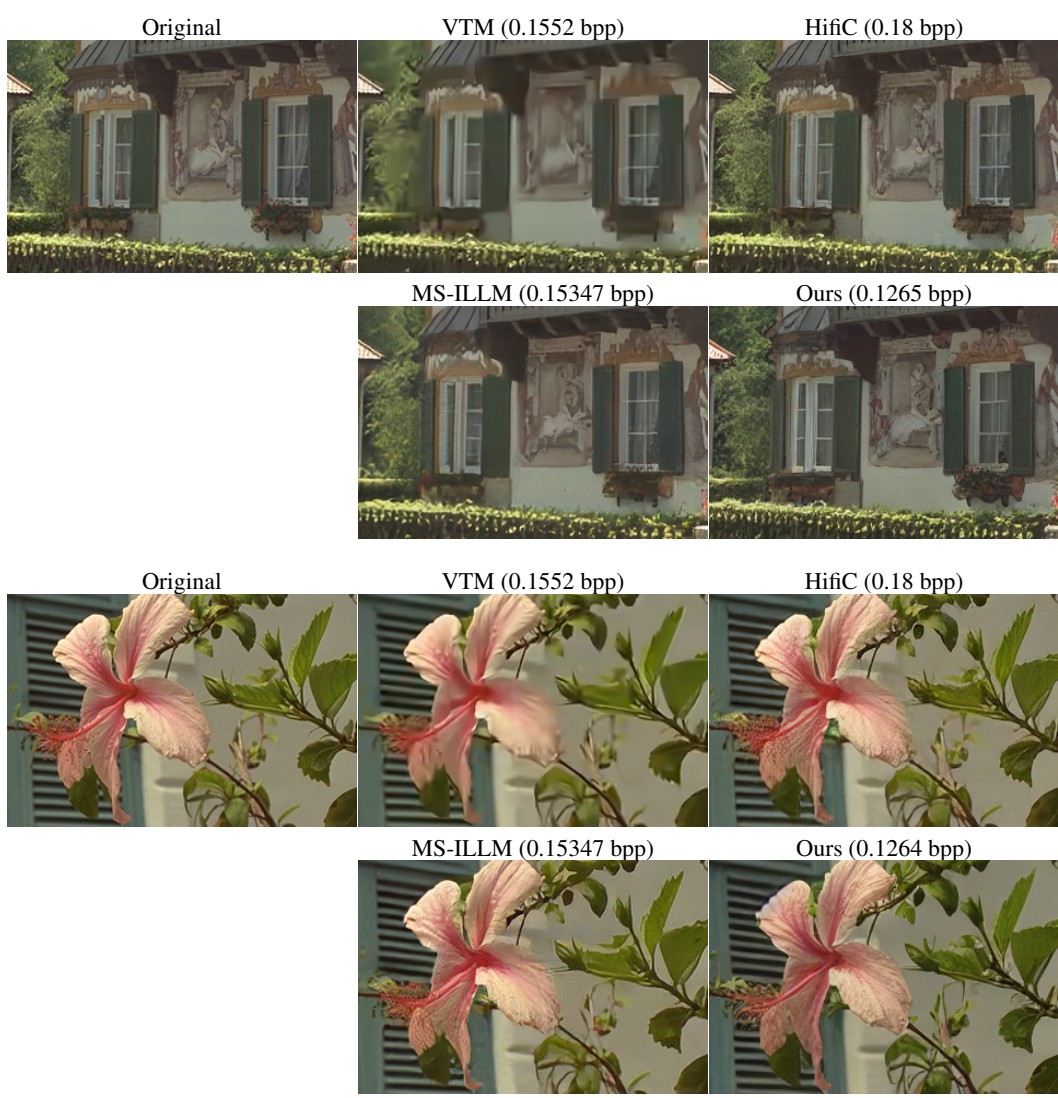

Figure 15: Reconstructions of Kodak images at the highest bitrate range we evaluated models on.

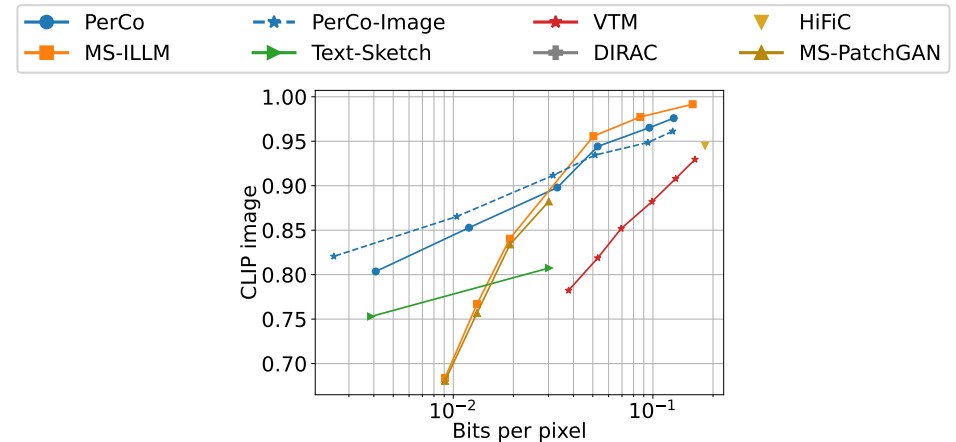

Figure 16: Evaluation in terms of the CLIP image metric.

