# OpenReview forum: "Towards image compression with perfect realism at ultra-low bitrates"
_ICLR.cc/2024/Conference — ICLR 2024 poster_

### Official Review · Reviewer_Z1Pi · 2023-10-17

**Soundness:** 3 good
**Presentation:** 3 good
**Contribution:** 2 fair
**Rating:** 6
**Confidence:** 4

**Summary:**

This paper proposes to use diffusion models as the decoder.
The diffusion decoder is conditioned on both a vector-quantized latent image representation and a textual image description (losslessly coded separately).
The proposed methods achieve realistic reconstructions at bitrates as low as 0.003 bits per pixel, significantly outperforming previous works.

**Strengths:**

1. The technical contribution is novel to me. I like the idea of conditioning the generative codec on a vector-quantized image representation along with a global textual image description to provide additional context.
2. The paper is well written and easy to follow. Enough background information is provided to general readers. It is a great work bridging learnt image compression and AIGC.
3. The results are significantly better than previous works.
4. Ablation studies are thorough and convincing.

**Weaknesses:**

1. Though the perceptual quality is SOTA, the decoding complexity is not compared. It is better to show the decoding complexity since diffusion models tend to be slower. Comparing the decoding complexity (i.e. latency and flops) with previous perceptual image compresion methods can provide more helpful information to the community.
2. Missing citations. Following Hoogeboom et al 2023, previous SOTA PO-ELIC should be compared or at least discussed if it is hard to make direct comparison: He, Dailan, et al. "PO-ELIC: Perception-Oriented Efficient Learned Image Coding." Proceedings of the IEEE/CVF Conference on Computer Vision and Pattern Recognition. 2022.
3. Other Suggestions. The idea of conditioning the generative codec on image representation along with an extra descriptor (which is losslessly coded) is very interesting. I noticed that the following work provides a theoretical perspective on this conditional coding setting and also explains that the perceptual quality measured by FID is not affected by bitrate, please consider discussing this to provide more information to readers.
Xu, Tongda, et al. "Conditional Perceptual Quality Preserving Image Compression." arXiv preprint arXiv:2308.08154 (2023).

**Questions:**

Overall I think this is a good work. Please refer to the weakness part for the discussion phase

---

> ### Author Response · Authors · 2023-11-14
>
> We thank the reviewer for their time and constructive comments and address the questions below:
>
> > Though the perceptual quality is SOTA, the decoding complexity is not compared. It is better to show the decoding complexity since diffusion models tend to be slower. Comparing the decoding complexity (i.e. latency and flops) with previous perceptual image compresion methods can provide more helpful information to the community.
>
> As stated in the joint answer, we performed tests for encode and decode speed and added a table with the results to the Appendix.
>
> > Missing citations. Following Hoogeboom et al 2023, previous SOTA PO-ELIC should be compared or at least discussed if it is hard to make direct comparison: He, Dailan, et al. "PO-ELIC: Perception-Oriented Efficient Learned Image Coding." Proceedings of the IEEE/CVF Conference on Computer Vision and Pattern Recognition. 2022.
>
> Thank you for pointing out to this relevant reference. In the related work section of the revised manuscript, we add a discussion on PO-ELIC which builds upon ELIC by adding a mixture of losses including adversarial ones to achieve better perceptual quality. They evaluate at bigger bitrates than us (their minimum bitrate is 0.075), and they do not provide code so it is difficult for us to include it in our benchmark. We already compare to three other adversarial-based codecs: HifiC as well as MS-ILLM and Multi-Realism which are more recent than PO-ELIC.
>
> > Other Suggestions. The idea of conditioning the generative codec on image representation along with an extra descriptor (which is losslessly coded) is very interesting. I noticed that the following work provides a theoretical perspective on this conditional coding setting and also explains that the perceptual quality measured by FID is not affected by bitrate, please consider discussing this to provide more information to readers. Xu, Tongda, et al. "Conditional Perceptual Quality Preserving Image Compression." arXiv preprint arXiv:2308.08154 (2023).
>
> Thank you for providing this relevant reference. This work introduces the notion of conditional perceptual quality, by extending perceptual quality to be conditioned on a specific side information. Given that we condition our generative codec on a losslessly encoded text description with the objective of achieving better preservation of text and local semantic information (measured by CLIP score/mIoU) and optimal image quality (FID), our work is very related to the idea of designing an optimal conditional perceptual quality preserving codec. We include this discussion in the related work.

---

> > ### Comment · Reviewer_Z1Pi · 2023-11-18
> > **Post rebuttal**
> >
> > Thanks for the reply, I keep my score

---

### Official Review · Reviewer_p6Yn · 2023-10-20

**Soundness:** 3 good
**Presentation:** 3 good
**Contribution:** 3 good
**Rating:** 6
**Confidence:** 4

**Summary:**

## Summary
* The authors present a technique to compress images into 0.003 bbp by text-to-image latent diffusion models. They show that in very low bitrate, pretrained caption becomes a very efficient latent representation for image compression.

**Strengths:**

## Strength
* There has been several diffusion based image compression works. A major problem of diffusion based compression is that what special advantage can diffusion brings with the cost of decoding time. This work find a specific scenario (ultra low bitrate) that is hard to be achieved by GAN models, which justifies the adoptation of diffusion over GAN, and I like it a lot.
* The visual results look strikingly good. No zoom-up is needed to see the obvious advantage over other methods.
* It is also interesting to see that text is a efficient representation for image compression.

**Weaknesses:**

## Weakness
* I am not really sure what is the point of evaluating LPIPS in Fig.3. Obviously MS-ILLM and HiFiC are trained with LPIPS, and this brings unfair advantage over the proposed approach. And as the authors are optimizing their approach towards divergence based perceptual quality [Blau 2018], using FID and KID should be enough. Reporting MS-SSIM and mIoU is also weird.
* The authors think achieveing a low MS-SSIM, PSNR and LPIPS is a weakness, which I can not agree. By RDP theory [Blau 2019], it is absolutely normal for perceptual codec being outperformed on image-wise distortion metrics.
* The authors are enouraged to present results in their original aspect ratio. Currently some results in Fig. 1 looks weird as they are squeezed horizontally.

**Questions:**

## Questions
* In MS-ILLM, the VQ-VAE is forzen once after pre-trained. While in this paper the VQ-VAE participate the end-to-end training. Then a natural question to ask is whether include the VQ-VAE of MS-ILLM in training improves the result?
* A recent preprint [Conditional Perceptual Quality Preserving Image Compression] also justifies the adoptation of side information in image compression. The authors can probably connect their work to rate-distortion-perception trade-off with the approach similar the in preprint.

---

> ### Author Response · Authors · 2023-11-14
>
> We thank the reviewer for their time and constructive comments and address the questions below:
>
> > I am not really sure what is the point of evaluating LPIPS in Fig.3. Obviously MS-ILLM and HiFiC are trained with LPIPS, and this brings unfair advantage over the proposed approach. And as the authors are optimizing their approach towards divergence based perceptual quality [Blau 2018], using FID and KID should be enough. Reporting MS-SSIM and mIoU is also weird.
>
> We agree with the reviewer that our goal is to design a perceptual codec and that FID/KID should be sufficient to evaluate the perceptual quality of our model. As we wanted to compare in the setting of RDP theory [Blau 2019], we included an evaluation on image similarity metrics LPIPS and MS-SSIM because they are widely used in the compression litterature. As rightly pointed out, LPIPS and MS-SSIM are not very meaningful in the low bitrate regime (see Fig. 14 in Appendix), so we included mIoU and CLIP score as they offer alternative ways of assessing distribution matching (mIoU having spatial localization and CLIP using a different image recognition mechanism). In the revised manuscript we further clarify the selection of metrics for evaluation.
>
> > The authors think achieveing a low MS-SSIM, PSNR and LPIPS is a weakness, which I can not agree. By RDP theory [Blau 2019], it is absolutely normal for perceptual codec being outperformed on image-wise distortion metrics.
>
> We agree with the reviewer. We include this discussion in 'Limitations' paragraph of the paper.
>
> > The authors are enouraged to present results in their original aspect ratio. Currently some results in Fig. 1 looks weird as they are squeezed horizontally.
>
> Thanks for the valuable remark. We update all figures in the paper with their original aspect ratio.
>
> > In MS-ILLM, the VQ-VAE is forzen once after pre-trained. While in this paper the VQ-VAE participate the end-to-end training. Then a natural question to ask is whether include the VQ-VAE of MS-ILLM in training improves the result?
>
> In our model, we use an autoencoder that maps RGB images to the latent space of the diffusion model. This autoencoder is pretrained and frozen when training our models and we train a small hyper-encoder in the latent space for vector quantization (see paragraph 'Implementation details' in Section 4.1). In MS-ILLM, the VQ-VAE is only used as a continuous-to-discrete mapping for the purpose of training an adversarial loss. Our model does not use an adversarial loss, so our assessment is the two are not comparable. If we misunderstood the reviewer’s intent, we could reconsider a rephrasing of this request.
>
> > A recent preprint [Conditional Perceptual Quality Preserving Image Compression] also justifies the adoptation of side information in image compression. The authors can probably connect their work to rate-distortion-perception trade-off with the approach similar the in preprint.
>
> Thank you for providing this relevant reference. This work introduces the notion of conditional perceptual quality, by extending perceptual quality to be conditioned on a specific side information. Given that we condition our generative codec on a losslessly encoded text description with the objective of achieving better preservation of text and local semantic information (measured by CLIP score/mIoU) and optimal image quality (FID), our work is very related to the idea of designing an optimal conditional perceptual quality preserving codec. We include this discussion in the related work.

---

> > ### Comment · Reviewer_p6Yn · 2023-11-16
> > **Thanks for your rebuttal**
> >
> > Thanks for your rebuttal, most of my concerns are addressed. The images look better in their original aspect ratio.
> >
> > p.s. the abstract is repeated twice in openreview.

---

### Official Review · Reviewer_FDrq · 2023-10-30

**Soundness:** 3 good
**Presentation:** 3 good
**Contribution:** 3 good
**Rating:** 6
**Confidence:** 4

**Summary:**

This paper proposes an image generative compression framework called PerCo. This model applies the diffusion model to the field of image compression and achieves visually good performance at extremely low bit rate compression. Compared to previous methods, this work leads to better visual quality as measured by subjective metrics like FID and KID.

**Strengths:**

The proposed method combines text and image description for the diffusion model to improve compression performance. Compared to previous works, it achieves higher quality and more realistic image representation at very low bit rates. The model proposed is valuable for future exploration of applying diffusion models to the field of image compression.
Moreover, the semantic logic and argumentation of this article are clear. The extensive comparative analysis also makes the article more convincing.

**Weaknesses:**

1）The training and inference complexity of such diffusion-based methods is usually large. The authors should provide some preliminary results like running time or memory consumption.

2）Another existing diffusion-based image compression framework [1] combining text and image information can be discussed.

[1] Pan, Zhihong, Xin Zhou, and Hao Tian. "Extreme Generative Image Compression by Learning Text Embedding from Diffusion Models." arXiv preprint arXiv:2211.07793 (2022).

**Questions:**

A minor question: on page 4, in “Local spatial encoding” paragraph line 6, “Third, we proceed to quantization of $H_s$ to obtain $z_g$ via vector quantization”.  From my understanding, $z_g$ should be $z_l$ ? (ps, a typo here: “quatization” to “quantization” )

---

> ### Author Response · Authors · 2023-11-14
>
> We thank the reviewer for their time and constructive comments and address the questions below:
>
> > The training and inference complexity of such diffusion-based methods is usually large. The authors should provide some preliminary results like running time or memory consumption.
>
> As stated in the joint answer, we performed tests for encode and decode speed and added a table with the results to the Appendix.
>
> > Another existing diffusion-based image compression framework [1] combining text and image information can be discussed. [1] Pan, Zhihong, Xin Zhou, and Hao Tian. "Extreme Generative Image Compression by Learning Text Embedding from Diffusion Models." arXiv preprint arXiv:2211.07793 (2022).
>
> Thanks for pointing out this relevant reference. In [1], they optimize a textual embedding on top of a pretrained text-to-image diffusion model. They also design a compression guidance method used at each denoising step during inference to better reconstruct original images. Their lowest compression rate is 0.042 bits per pixel which is lower than most of the compression baselines, but still much higher than 10x our minimum bitrate of 0.0031 bits per pixel. We include this paper in the related work.
>
> > A minor question: on page 4, in “Local spatial encoding” paragraph line 6, “Third, we proceed to quantization of $H_s$
>  to obtain $z_g$ via vector quantization”. From my understanding, $z_g$ should be $z_l$ ? (ps, a typo here: “quatization” to “quantization” )
>
> Thanks a lot for pointing out the two typos, we correct them in the revised manuscript.

---

> > ### Comment · Reviewer_FDrq · 2023-11-20
> >
> > Thanks for the response. I will keep my rating.

---

### Official Review · Reviewer_6W37 · 2023-11-07

**Soundness:** 3 good
**Presentation:** 3 good
**Contribution:** 2 fair
**Rating:** 6
**Confidence:** 4

**Summary:**

This paper proposes a new image compression method, which decodes compressed latent features with a iterative diffusion model, instead of a single-pass feed-forward decoder. Specifically, the introduced diffusion compressor conditions on two kinds of features:  (1) a global textual image description obtained from a image captioning model; (2) a vector-quantized image representation in latent space.  According to the experimental results, the proposed method beats the state-of-the-art methods in visual quality at ultra-low bitrate.

**Strengths:**

- This paper introduces a new image compression architecture. It proposes to combine a global textual image description and a vector-quantized latent image representation as the conditions of the latent diffusion model to decode the images.
- The reconstruction results are much better than the exisiting works at ultra-low bitrates.

**Weaknesses:**

- The idea of applying diffusion model on image compression is not new (as reviewed in the paper). But at least for me, combining it with textual information is interesting and motivating for image compression task.
- The proposed method only optimizes the distortion/perceptual term but drops the rate term. It is not justified why the rate term can be dropped.
What impact will have if include the rate term into optimization?
- One limitation of diffusion model is the inference speed. This paper do not provide the running time of the proposed method, especifically the decoding time. A fair comparsion of running time (both encoding and decoding ) with the competing methods (both learned and traditional) should be provided and discussed.

**Questions:**

Please see the weaknesses.

---

> ### Author Response · Authors · 2023-11-14
>
> We thank the reviewer for their time and constructive comments and address the questions below:
>
> > The proposed method only optimizes the distortion/perceptual term but drops the rate term. It is not justified why the rate term can be dropped. What impact will have if include the rate term into optimization?
>
> In our approach, the rate is controlled through two hyper-parameter settings: the spatial resolution of the hyper-latents, and the vocabulary size. Therefore, the rate can be considered fixed when training our models, and therefore dropped from the objective function. To clarify this, we rephrase the first paragraph of section 3.1.
>
> > One limitation of diffusion model is the inference speed. This paper do not provide the running time of the proposed method, especifically the decoding time. A fair comparsion of running time (both encoding and decoding ) with the competing methods (both learned and traditional) should be provided and discussed.
>
> As stated in the joint answer, we performed tests for encode and decode speed and added a table with the results to the Appendix.

---

> > ### Comment · Reviewer_6W37 · 2023-11-22
> >
> > Thanks for your reply. My concerns are well addressed. I still stick with my rating.

---

### Author Response · Authors · 2023-11-14
**General Answer**

We would like to thank the reviewers for their valuable feedback, and are encouraged by their positive reception of our work:
- (6W37) "results are much better than the exisiting works" ,
- (FDrq) "valuable for future exploration of applying diffusion models to the field of image compression",
- (p6Yn) "results look strikingly good",
- (Z1Pi) "significantly outperforming previous works".

We respond below to specific points raised by each reviewer, and include a single response about inference speed as requested by multiple reviewers.

We have updated the paper. Revision parts are highlighted with red text and summarized as follows:
- (6W37, FDrq, Z1Pi) We include the table and discussion about inference speed measurements in the Appendix.
- (Z1Pi, p6Yn, FDrq) We added a discussion about the papers "Conditional Perceptual Quality Preserving Image Compression", "PO-ELIC: Perception-Oriented Efficient Learned Image Coding" and "Extreme Generative Image Compression by Learning Text Embedding from Diffusion Models" in the Related Work.
- (6W37) We rephrase the first paragraph of section 3.1 to better explain why we are dropping the rate term.
- (FDrq) We correct the two typos on page 4 in "Local spatial encoding" paragraph line 6.
- (p6Yn) We further clarify the selection of evaluation metrics in the paragraph 'Metrics' of Section 4.1. We update discussion about poorer image reconstruction scores in the 'Limitations' section of the paper.
- (p6Yn) We update all the Figures in the paper with original aspect ratio.

We hope we addressed all the reviewers' concerns, and we will be happy to provide additional clarifications upon request.

---

### Author Response · Authors · 2023-11-14
**Joint answer about inference speed**

We agree with the reviewers that it would be interesting to include measurements of encoding and decoding speed of our model and others to the paper.

- We run benchmarking on A100 gpus using 20 denoising steps, and 5 denoising steps for bitrates higher than 0.05 bits per pixel.
We compute the runtime for all the Kodak images at resolution 512x768 and provide the average time in seconds with its standard deviation.
- The encoding complexity of PerCo is quite similar to adversarial-based compression methods such as MS-ILLM and HifiC (which has the same architecture as MS-ILLM). Encoding with Text-Sketch (PIC/PICS) is orders of magnitude slower because it is optimizing a text embedding for each image.
- The decoding complexity of PerCo using 20 denoising steps (bitrates below 0.05 bpp) is comparable to that of Text-Sketch (PIC) and about 8x faster than Text-Sketch (PICS), because PICS is conditioning on sketches which takes time to decompress. When using only five denoising steps, the decoding complexity drops rouhgly by a factor four, showing that the decoding is dominated by evaluations of the denoising network. Compared to HiFiC and MS-ILLM, the PerCo decoder using 5 denoising steps is about eight times slower, and when using 20 steps this becomes a factor 32.
- Please note that the running time of VTM is computed on CPU, so it may explain the longer runtime.
- About the diffusion-based models DIRAC and HFD/DDPM, we couldn't evaluate their models as there is no available codebase, but we found some evaluations in DIRAC paper about decoding time. They report a decoding time which is about four times slower than what they report for HifiC, which also shows that diffusion-based methods are slower than adversarial-based ones.
- We include these inference speed measurements and their discussion in the supplementary material of the revised manuscript.

| Kodak Images     | Encoding Speed (in sec.) | Decoding Speed (in sec.) |
| -------- | :------: | :------: |
VTM | 16.8924 $\pm$  7.5744 | 0.1352 $\pm$  0.0024 |
MS-ILLM | 0.0841 $\pm$  0,0211 |	0.0804 $\pm$  0,0068 |
Text-Sketch (PIC) |  163.0699 $\pm$  0.3795	| 2,7250 $\pm$  0,01162 |
Text-Sketch (PICS) | 190.2310 $\pm$  2,4758 | 19,2878 $\pm$  0,2511|
PerCo - 5 denoising steps | 0.0801 $\pm$  0.0183  | 0.6653 $\pm$  0.0092 |
PerCo - 20 denoising steps | 0.0801 $\pm$ 0.0183 | 2,5510 $\pm$ 0.0183 |

---

### Author Response · Authors · 2023-11-21
**Deadline for discussion period approaching**

Dear reviewers,

as the end of the discussion period is approaching, we would like to thank you again for the valuable feedback. Please let us know if the changes to the paper address your concerns. We are eager to engage in any further discussions if needed.

---

### Meta-Review · Area_Chair_PGCp · 2023-12-10

**Metareview:**

The authors propose to use an iterative diffusion model to decompress latent features in a new image compression method.
The model is conditioned on a vector-quantized image representation in latent space and on a global textual image description obtained from an image captioning model. The proposed method provides state-of-the-art visual quality at ultra-low bitrates.

The four reviewers and the meta-reviewer are generally appreciating the proposed compression method and the results achieved in both quantitative and qualitative evaluations when compared with prior works.

Weaknesses are derived from limited novelty at the level of techniques and concepts, as well as the time complexity / decoding complexity inherited from the adoption of an iterative diffusion model.

The authors provided responses to all the reviewers' concerns and the reviewers are generally unanimously leaning towards acceptance of the paper (6,6,6,6).

The meta-reviewer after carefully reading the reviews, the discussions, and the paper, agrees with the reviewers and recommends acceptance.

**Justification For Why Not Higher Score:**

While the idea and the effectiveness of the method are strengths, the advances at the machine learning or theoretical level are limited and the interest for the addressed topic is rather limited as well within the ICLR community.

**Justification For Why Not Lower Score:**

Four reviewers and the meta-reviewer agree that the paper has merits, and the contributions are sufficient and of interest. There is no significant flaw to impede publication.

---

### Decision · Program_Chairs · 2024-01-16

Accept (poster)